# (GIGA)byte

DATA RELEASE

# Genome assembly and annotation of the tambaqui (*Colossoma macropomum*): an emblematic fish of the Amazon River Basin

Alexandre Wagner Silva Hilsdorf[1,*,†], Marcela Uliano-Silva[2,†], Luiz Lehmann Coutinho[3], Horácio Montenegro[3], Vera Maria Fonseca Almeida-Val[4] and Danillo Pinhal[5,*]

1 Integrated Center of Biotechnology, University of Mogi das Cruzes, P.O. Box 411, Mogi das Cruzes, SP 08780-911, Brazil
2 Wellcome Sanger Institute, Saffron Walden, Hinxton, Cambridgeshire CB10 1SA, UK
3 Animal Science Department, University of São Paulo (USP)/Luiz de Queiroz College of Agriculture (ESALQ), Piracicaba, SP 13418-900, Brazil
4 Brazilian National Institute for Research of the Amazon, Laboratory of Ecophysiology and Molecular Evolution, Manaus, AM 69067-375, Brazil
5 Department of Chemical and Biological Sciences, Institute of Biosciences of Botucatu, São Paulo State University (UNESP), Botucatu, SP 18618-689, Brazil

**Submitted:** 20 July 2021

* Corresponding authors. E-mail: wagner@umc.br; danillo.pinhal@unesp.br

† Contributed equally.

Preprint submitted at https://www.biorxiv.org/content/10.1101/2021.09.08.459456v1

## ABSTRACT

*Colossoma macropomum*, known as "tambaqui", is the largest Characiformes fish in the Amazon River Basin and a leading species in Brazilian aquaculture and fisheries. Good quality meat and excellent adaptability to culture systems are some of its remarkable farming features. To support studies into the genetics and genomics of the tambaqui, we have produced the first high-quality genome for the species. We combined Illumina and PacBio sequencing technologies to generate a reference genome, assembled with 39× coverage of long reads and polished to a consensus quality value (QV) of 36 with 130× coverage of short reads. The genome was assembled into 1269 scaffolds (a total of 1,221,847,006 bases), with a scaffold N50 size of 40 Mb, where 93% of all assembled bases were placed in the largest 54 scaffolds corresponding to the diploid karyotype of the tambaqui. Furthermore, the NCBI Annotation Pipeline annotated genes, pseudogenes, and non-coding transcripts using the RefSeq database as evidence, guaranteeing a high-quality annotation. A Genome Data Viewer for the tambaqui was produced, which will benefit groups interested in exploring the unique genomic features of the species. The availability of a highly accurate genome assembly for tambaqui provides the foundation for the discovery of novel ecological and evolutionary insights, and is a helpful resource for aquaculture.

**Subjects** Genetics and Genomics, Evolutionary Biology, Marine Biology

## INTRODUCTION

The Amazon Basin harbors enormous freshwater ichthyo diversity throughout its rivers and tributaries, with 2406 validated freshwater native fish species from 232,936 georeferenced records [1]. *Colossoma macropomum* (NCBI:txid42526, fishbase ID:263) is the largest Characiformes representative found across the Amazon River and its tributaries, with individuals reaching 1 meter in length and 30 kg in weight (Figure 1) [2]. This species is known by different common names, such as "tambaqui" in Brazil and "cachama Negra" in

**Figure 1.** *Colossoma macropomum* individual used for whole genome sequencing.

Colombia. Tambaquis are omnivore/frugivore benthopelagic fish, and they have an essential ecological role as seed dispersers [3]. They are potamodromous fish, with upstream migration and reproduction taking place in the white waters along woody shores between November and February [4]. The tambaqui is an important food and income source for Amazonian fishing communities; it is the most frequently farmed native fish species in Brazil, with a production of 101,079 metric tons in 2019 [5, 6].

The ecological and economic importance of the tambaqui means it is a comparatively well-studied species. Research to date has focused on its biological adaptations to the Amazon River waters, and on the genetics of production traits to assist selective breeding programs. Transcriptomic characterization of tambaqui exposed to (i) distinct climate change scenarios, and (ii) during gonadal differentiation, has provided helpful resources for understanding the molecular mechanisms underlying both adaptation to a future new climate and the process of sex determination [7–9]. Other molecular mechanisms related to enzymatic capacity for long-chain polyunsaturated fatty acid biosynthesis have also been confirmed by the functional characterization of core genes in these processes [10, 11]. The first steps for deciphering the structure and functional dynamics of the tambaqui genome have already been taken, with large-scale single nucleotide polymorphism (SNP) discovery allowing a high-density genetic linkage map of the species to be built [12], along with preliminary microRNA identification and characterization [13]. Equally pertinent are the new findings in morphology: specimens lacking intramuscular bones were identified in a fish farm in Brazil; however, the genetic and molecular mechanisms underlying the expression of such desirable phenotypes for the fish market remain unknown [14, 15].

Considering the great need for increased genetic resources for the tambaqui to assist fishery management and aquaculture [16], here we present the first high-quality reference genome for *C. macropomum*. This complete set of DNA provides a valuable resource for the study of evolutionary and functional genomics in bony fishes, providing a window of opportunity to reveal singularities of the tambaqui genome, as well as to help develop molecular techniques to improve selective breeding programs.

**Table 1.** NCBI accessions for the genes used for taxonomic ascertainment.

| Genes | COI | TROP | fkh | RAG2 | sina |
|---|---|---|---|---|---|
| Species | | | | | |
| *Colossoma macropomum* | HQ420845.1 | HQ420888.1 | AY817328.1 | AY804061.1 | AY790059.1 |
| *Piaractus brachypomus* | HQ420838.1 | HQ420883.1 | AY817392.1 | AY804112.1 | AY790125.1 |
| *Piaractus mesopotamicus* | HQ420837.1 | HQ420878.1 | AY817398.1 | AY804118.1 | AY790131.1 |

COI: Cytochrome c oxidase I; TROP: Alpha tropomyosin; fkh: fork head domain protein; RAG2: recombination activating gene 2; sina: absentia 1A.

**Table 2.** Final statistics for the genome assembly of *Colossoma macropomum*.

| | Contig length (bp) | Scaffold length (bp) | Number of Contigs | Number of Scaffolds |
|---|---|---|---|---|
| Total | 1,221,809,066 | 1,221,847,006 | 1687 | 1269 |
| Max | 26,165,397 | 63,817,184 | — | — |
| N50 | 5,645,235 | 40,163,545 | 54 | 14 |
| N90 | 655,952 | 2,856,822 | 300 | 33 |

## METHODS

### DNA isolation and taxonomy identification

Genomic DNA was isolated from caudal fin-clip samples from a *C. macropomum* specimen obtained from the germplasm bank maintained by the National Center for Research and Conservation of Freshwater Aquatic Biodiversity of the Brazilian Ministry of the Environment. The specimen was a 3.5-kg female (Figure 1). To confirm the taxonomic status of the specimen used in this work, we carried out (i) an external morphological evaluation [17], and (ii) a preliminary genetic analysis of an initial Illumina run for *C. macropomum* using the *k*-mer-matching tool Seal from the BBTools package (v 37.90, RRID:SCR_016968) [18]. We downloaded the sequences of one mitochondrial and four nuclear genes of *C. macropomum* and its two close relatives, *Piaractus brachypomus* and *P. mesopotamicus* (Table 1). Then, we used Seal to ascertain the number of reads with exclusive *k*-mers matching the sequences of each species. Out of 264,813,582 reads, 1278 matched *C. macropomum*, 62 matched *P. brachypomus*, and none matched *P. mesopotamicus*, confirming the identity of the samples.

### Sequencing and assembly

Different data types were produced for the genome assembly of *C. macropomum*. High-molecular-weight DNA was extracted from muscle and fin clip using the MagMAX CORE nucleic acid purification kit (Thermo Fisher Scientific, Carlsbad, CA, USA) to produce PacBio continuous long reads (CLR) and Illumina paired and jumping reads (Table 2). The produced libraries were sequenced with both PacBio Single Molecule Real-Time (SMRT) sequencing technology using the Sequel system (RRID:SCR_017989) and four SMRT cells at RTL Genomics (Texas, USA), and with Illumina Hiseq2500 V4 equipment (RRID:SCR_016383) at the Functional Genomics Core Facility, Esalq-USP (São Paulo, Brazil). Illumina read quality was checked with FastQC (RRID:SCR_014583) [19] and trimmed for adaptors and low-quality bases with BBDuk (BBTools 37.90; SW15-20, RRID:SCR_016969). Genome size and heterozygosity were estimated by *k*-mer (*k* = 21) analysis (Figure 2A), performed with the sequenced Illumina data using meryl kmer counter (v1.3) [20], implemented in Canu assembler [21] and GenomeScope [22].

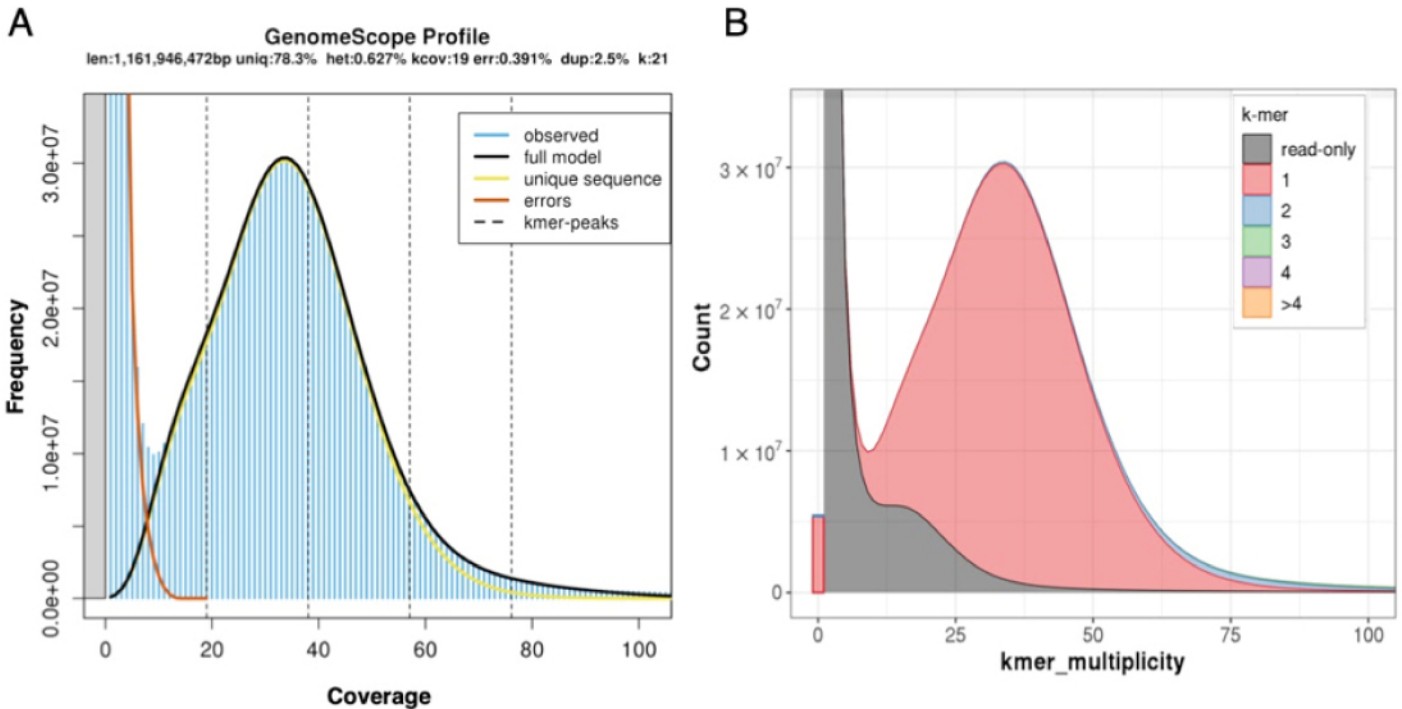

**Figure 2.** (A) *k*-mer composition of sequenced short Illumina reads (Table 3) of the tambaqui *Colossoma macropomum*. (B) A merqury *k*-mer analysis of the final tambaqui genome bases against its sequenced Illumina reads.

**Table 3.** Summary of genome sequencing data generated using multiple sequencing technologies. Sequencing coverage was based on the estimated genome size (1.16 Gbp) generated for *Colossoma macropomum* by *k*-mer analysis (*k* = 21) of the Illumina sequencing data.

| Library type | Insert size (bp) | Raw data (Gbp) | Clean data (Gbp) | Average read length (bp) | N50 read length (bp) | Clean data sequencing coverage (×) |
|---|---|---|---|---|---|---|
| Illumina (R1 and R2) | 400 | 59.08 | 52.93 | 100 | — | 44.89 |
| Illumina (R1 and R2) | 4000 | 78.81 | 57.69 | 81 | — | 49.7 |
| Illumina (R1 and R2) | 8000 | 55.59 | 41.31 | 83 | — | 35.6 |
| Pacbio CLR | — | 45.58 | — | 9749 | 17667 | 39.2 |
| Total | | | | | | 169.39 |

CLR: continuous long reads.

The 21-mer distribution of the Illumina data obeyed the theoretical Poisson distribution (Figure 2A). The genome size was estimated as 1.16 Gbp (gigabase pairs) with heterozygosity of 0.62%. Based on these estimations, we sequenced a 39× coverage of the tambaqui genome in long PacBio reads, and 130× in short Illumina reads (Table 3). For the genome assembly, PacBio reads were input to the assembler Flye (v2.5, RRID:SCR_017016) [23] with the parameters "genome-size 1.5g - pacbio-raw". Then, the assembly was polished using the Illumina reads with Pilon software (RRID:SCR_014731) [24], and the parameters "frags" for paired reads and "jumps" for mate-pair reads. Finally, the assembly of the tambaqui had one round of purging with Purge_Dups (RRID:SCR_021173) [25]. Purging was performed to remove any sequences representing duplicated portions of a chromosome, which can be erroneously kept in assemblies when the divergence level of those regions in both haplotypes is high. This removed 1,167 contigs and 26 Mbp (megabase pairs) of haplotypic

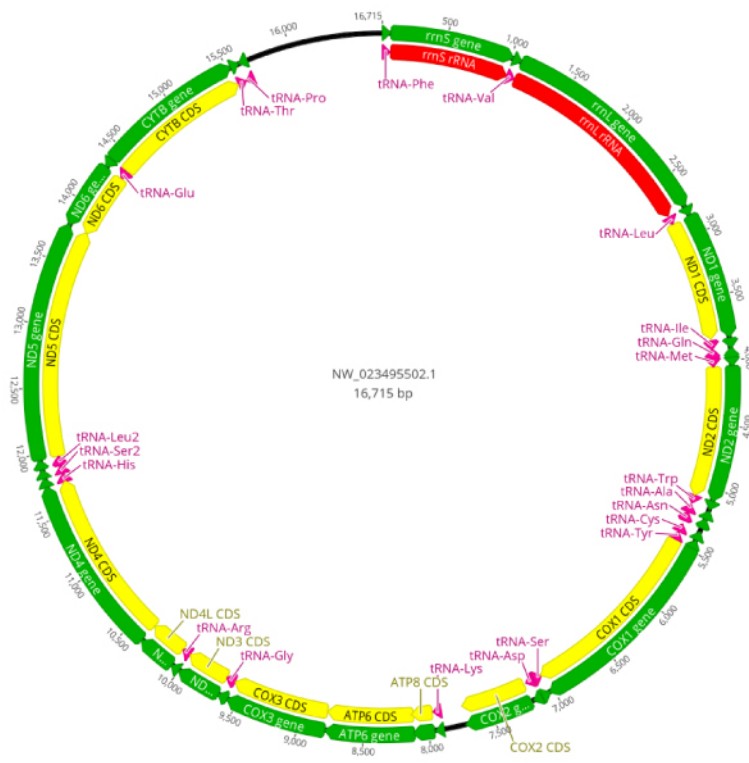

**Figure 3.** Mitogenome of *Colossoma macropomum*.

retention. The final tambaqui genome was assembled into 1,269 scaffolds with a scaffold N50 of 40 Mbp and a total assembly length of 1,221,847,006 bp (Table 2). A fraction of 93% of the genome is assembled on 54 scaffolds, which represent the main tambaqui karyotype [26]. We have also identified the mitochondrial genome (Figure 3) within our assembled genome: it is represented by scaffold NW_023495502.1, which is 16,715 bp in length and has conserved gene content and synteny with the *C. macropomum* mitogenome available at the National Center for Biotechnology Information (NCBI; KP188830.1).

## Repeat sequences and gene annotation

We identified repeat sequences in *C. macropomum* using homology-based, and *de novo* approaches. A *de novo* library of repeats was created for the tambaqui using RepeatModeler2 package (RRID:SCR_015027) [27]. This library was then combined with RepBase (release 26.04, RRID:SCR_021169) [28], forming the final "teleost" library with which *C. macropomum* genome repeats were searched. Table 4 presents the repeat summary of *C. macropomum*: 52.49% of the genome comprises repeats, of which 49.78% are interspersed repeats. The *C. macropomum* genome was submitted to NCBI for annotation. The robust NCBI Eukaryotic Annotation Pipeline uses homology-based and *ab initio* gene prediction to annotate genes (including protein-coding and non-coding, such as lncRNAs and snRNAs), pseudogenes, transcripts, and proteins. Details of the pipeline are described in the NCBI Annotation HandBook [29]. Briefly: first, repeats are masked with RepeatMasker (RRID:SCR_012954) [30] and Window Masker [31]. Subsequently, transcripts, proteins, and RNA-seq data from the NCBI database are aligned to the genome with

**Table 4.** Repeat annotation. Annotation of repeats done for *Colossoma macropomum* with a *de novo* library built with RepeatModeler added to a Repbase teleost library. The final library was used as input for RepeatMasker.

| Bases masked: 641,307,197 bp (52.49%) | | Number of elements* | Length occupied | % of sequence |
|---|---|---|---|---|
| Retroelements | | 131365 | 35210915 | 2.88 |
| | SINEs: | 3369 | 162823 | 0.01 |
| | Penelope | 2614 | 206056 | 0.02 |
| | LINEs: | 88299 | 25531727 | 2.09 |
| | CRE/SLACS | 5 | 195 | 0 |
| | L2/CR1/Rex | 54941 | 16069764 | 1.32 |
| | R1/LOA/Jockey | 1613 | 158940 | 0.01 |
| | R2/R4/NeSL | 688 | 137427 | 0.01 |
| | RTE/Bov-B | 9260 | 3512602 | 0.29 |
| | L1/CIN4 | 9819 | 2801917 | 0.23 |
| | LTR elements: | 39697 | 9516365 | 0.78 |
| | BEL/Pao | 1824 | 655410 | 0.05 |
| | Ty1/Copia | 3452 | 196980 | 0.02 |
| | Gypsy/DIRS1 | 17593 | 6224074 | 0.51 |
| | Retroviral | 13302 | 1948492 | 0.16 |
| DNA transposons | | 428117 | 94637950 | 7.75 |
| | hobo-Activator | 50751 | 5464720 | 0.45 |
| | Tc1-IS630-Pogo | 270090 | 78887086 | 6.46 |
| | PiggyBac | 3206 | 517597 | 0.04 |
| | Tourist/Harbinger | 4980 | 440554 | 0.04 |
| | Other (Mirage, P-element, Transib) | 1414 | 117503 | 0.01 |
| Rolling circles | | 9904 | 2012553 | 0.16 |
| Unclassified: | | 2468233 | 478402494 | 39.15 |
| Total interspersed repeats | | | 608251359 | 49.78 |
| Small RNA: | | 2641 | 167105 | 0.01 |
| Satellites: | | 15326 | 2676106 | 0.22 |
| Simple repeats: | | 435230 | 23721925 | 1.94 |
| Low complexity | | 51965 | 4532860 | 0.37 |

*Most repeats fragmented by insertions or deletions have been counted as one element.

Splign [32] and ProSplign [33]. Those alignments are submitted to Gnomon [34] for gene prediction. Gnomon (i) merges non-conflicting alignments into putative models, then (ii) extends predictions missing a start and a stop codon or internal exon(s) using a hidden Markov model (HMM) algorithm. Finally, Gnomon (ii) builds pure *ab initio* predictions where it finds open reading frames of sufficient length but with no supporting alignment detected. Models built on RefSeq transcript alignments are given preference over overlapping Gnomon models with the same splice pattern. Table 5 presents a summary of the annotation of *C. macropomum*. A detailed description of the tambaqui genome annotation can be found on the NCBI Eukaryotic Annotation Page [35].

## RESULTS AND DISCUSSION

All sequencing data are available at the NCBI under the BioProject PRJNA702552, via Sequence Read Archive (SRA) accession numbers SRX10122091 to SRX10122101. The assembled genome and sequence annotations are available at the NCBI with the accession number GCF_904425465.1. The genome sequence and the annotation files—including CDS and proteins—can be downloaded from the NCBI FTP server [37]. Finally, a genome DataViewer was built for the tambaqui [38]. This browser is ideal for

**Table 5.** Summary of the annotated features of *Colossoma macromapum* genome.

| Feature | *Colossoma macropomum* |
|---|---|
| Genes and pseudogenes | 31,149 |
| Protein-coding | 26,670 |
| Non-coding | 3279 |
| CDSs | |
| Fully-supported | 43,938 |
| With >5% ab initio | 1648 |
| Partial | 267 |
| Protein assigned RefSeq(XP_) | 43,618 |
| Mean CDS length (bp) | 2011 |
| Mean intron length (bp) | 2631 |
| Mean exon length (bp) | 280 |
| Mean exon per gene | 12.02 |

Detailed annotation report can be found at [36].

further exploration of the tambaqui genome, especially by those who are not specialist bioinformaticians, such as geneticists working on selective breeding programs.

## Evaluating the completeness of the genome assembly and annotation

The final assembly of the tambaqui is 1.2 Gbp with a scaffold N50 size of 40.163 Mbp (Table 2). Figure 2A shows the DNA *k*-mer prediction of genome size done using the Illumina reads produced to polish this assembly. Further, Figure 2B presents a merqury [39] *k*-mer plot of the final assembly: merqury produces a mapping-free evaluation of *k*-mer completeness in genomes by comparing the assembly *k*-mers with raw reads for the specimen. In this case, we used the high-quality Illumina reads (Table 3) to plot the merqury evaluation against the genome *k*-mers. Figure 2B shows that (i) the *k*-mers in the genome are in accordance with its Illumina read *k*-mers, (ii) the assembly *k*-mers have the same distribution of the raw reads *k*-mer (2A), and that (iii) most of the assembly *k*-mers (pink color) are unique in the genome, showing that the final assembly of the tambaqui has low levels of haplotypic retention (blue color). The final phred-like merqury QV score is 36.73 (QV = 36. 73), meaning that the tambaqui assembled bases are more than 99.9% accurate. The merqury completeness score shows that 89.31% of kmers in the Illumina reads are present in the assembly, which is a good recovery of *k*-mers for a species with 0.6% heterozygosity.

For the tambaqui genome, 93% of the assembled bases are present in the largest 54 scaffolds. We performed a first nucleotide synteny analysis of Benchmarking Universal Single-Copy Ortholog (BUSCO) genes found in the first 54 scaffolds of *C. macropomum* against the BUSCO genes on genome of *Ictalurus punctatus* [40] using busco2fasta [41] and Circos [42]. The synteny is presented in Figure 4. *C. macropomum* and *I. punctatus* shared a common ancestor ~150 million years ago [43]. The image shows a good degree of synteny in terms of BUSCO genes; for a number of times entire chromosomes are syntenic. Figures 5 and 6 show similar analysis with *C. auratus* [44] and *Astyanax mexicanus* [45] of different levels of relatedness to *C. macropomum*, demonstrating the potential of this highly contiguous genome for studies of chromosome evolution.

Finally, we performed a general gene presence analysis of the *C. macropomum* genome using BUSCO software (v5.0.0, RRID:SCR_015008) [46] and the OrthoDB

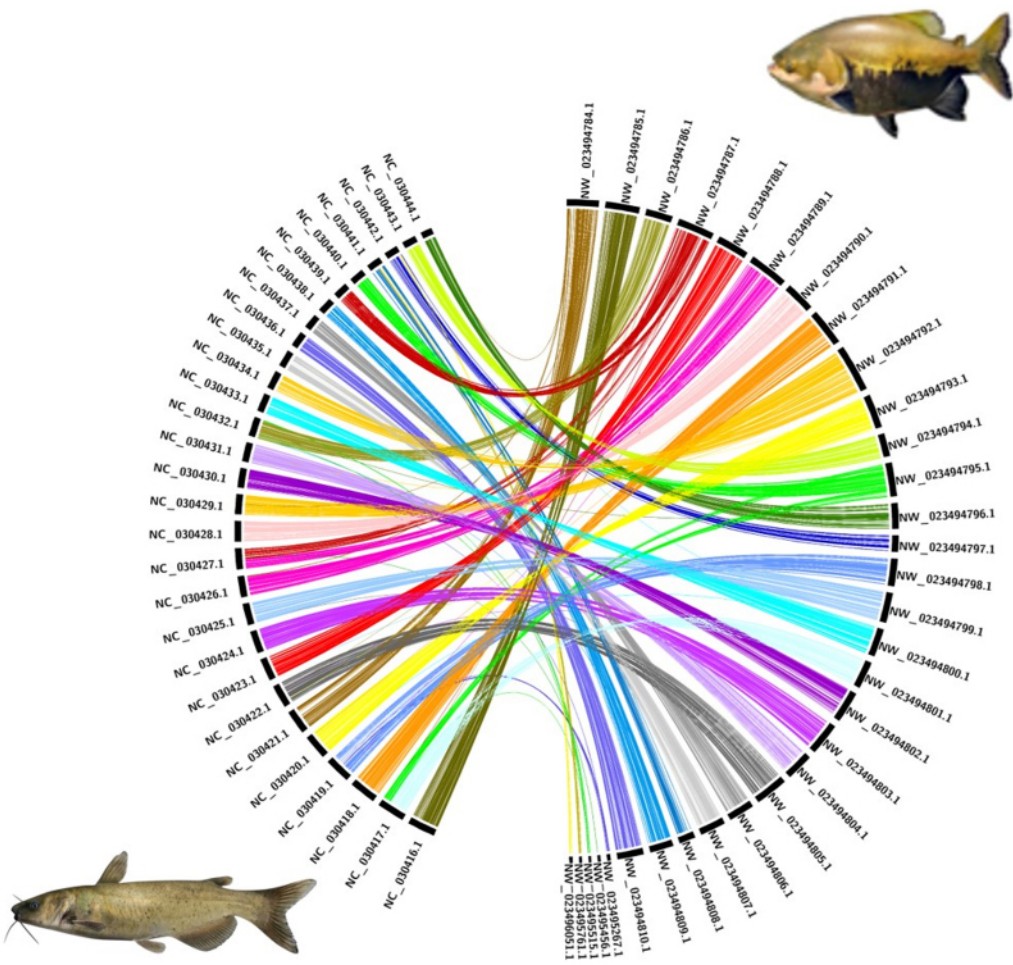

**Figure 4.** BUSCO gene synteny of *Colossoma macropomum* (tambaqui; right) and *Ictalurus punctatus* (channel catfish; left). Synteny analysis of single copy genes reveals low conservation of homologous gene order between the species. Most *C. macropomum* genes are pulverized into several linkage groups of the *I. punctatus* genome, which may reflect the different genome evolutionary events experienced by these species.

(RRID:SCR_011980) [47] library actinopterygii_odb10. BUSCO v5 recovered 96.5% of complete BUSCO genes out of 3640 genes searched, where 95.5% were complete and single copy, 1.0% were duplicated, 1.0% were fragmented, and 2.5% were missing. Once again, this demonstrates the quality of the tambaqui assembly.

## Gene family identification and phylogenetic analysis of *C. macropomum*

To identify gene families among *C. macropomum* and other species, we downloaded the whole genome predicted protein sequences from the NCBI Eukaryotic Annotation Page of *Oreochromis niloticus* (GCF_001858045.2), *Carassius auratus* (GCF_003368295.1), *Danio rerio* (GCF_000002035.6), *Latescalcarifer* (GCF_001640805.1), *Cyprinus carpio* (GCF_000951615.1), *Rhincodontypus* (GCF_001642345.1), *Poecilia formosa* (GCF_000485575.1), *Ictalurus punctatus* (GCF_001660625.1), *Astyanax mexicanus* (GCF_000372685.2),

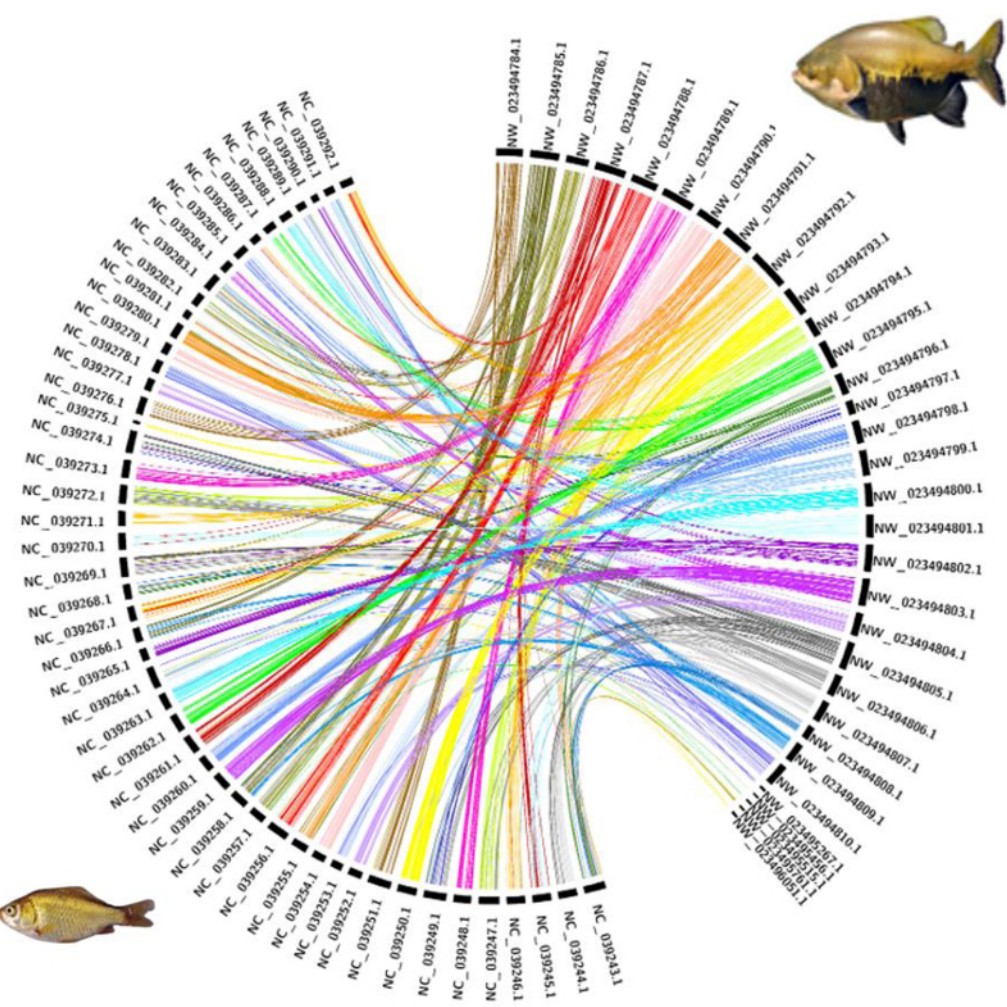

**Figure 5.** BUSCO gene synteny of _Colossoma macropomum_ (tambaqui; right) and _Carassius auratus_ (goldfish; left). Synteny analysis of single copy genes reveals low conservation of homologous gene order between the species. Most _C. macropomum_ genes are pulverized into several linkage groups of the _C. auratus_ genome, reflecting the different genome evolutionary events experienced by these species.

_Oncorhynchus mykiss_ (GCF_013265735.2) and _Pygocentrus nattereri_ (GCF_001682695.1). We then input this data to Orthofinder (v2.5.2) [48]. Of all the proteins imputed from the 12 species, Orthofinder assigned 97.3% of the proteins to 31,794 orthogroups. There were 10,939 orthogroups with all the species present, and 33 of them consisted of single-copy genes. Those 33 single-copy orthologs were used to generate a phylogeny (Figure 7). First, single-copy genes were aligned with MAFFT (v7.455, RRID:SCR_011811) [49], and alignments were trimmed with trimAL (v1.4. rev15, RRID:SCR_017334) [50]. Then, a supermatrix was created using geneStitcher.py [51], which was imputed to PhyML (RRID:SCR_014629) [52] for a phylogeny with the amino acid substitution model LG and 100 bootstrap replicates. The phylogeny presented herein (Figure 7) is consistent with other studies [53, 54].

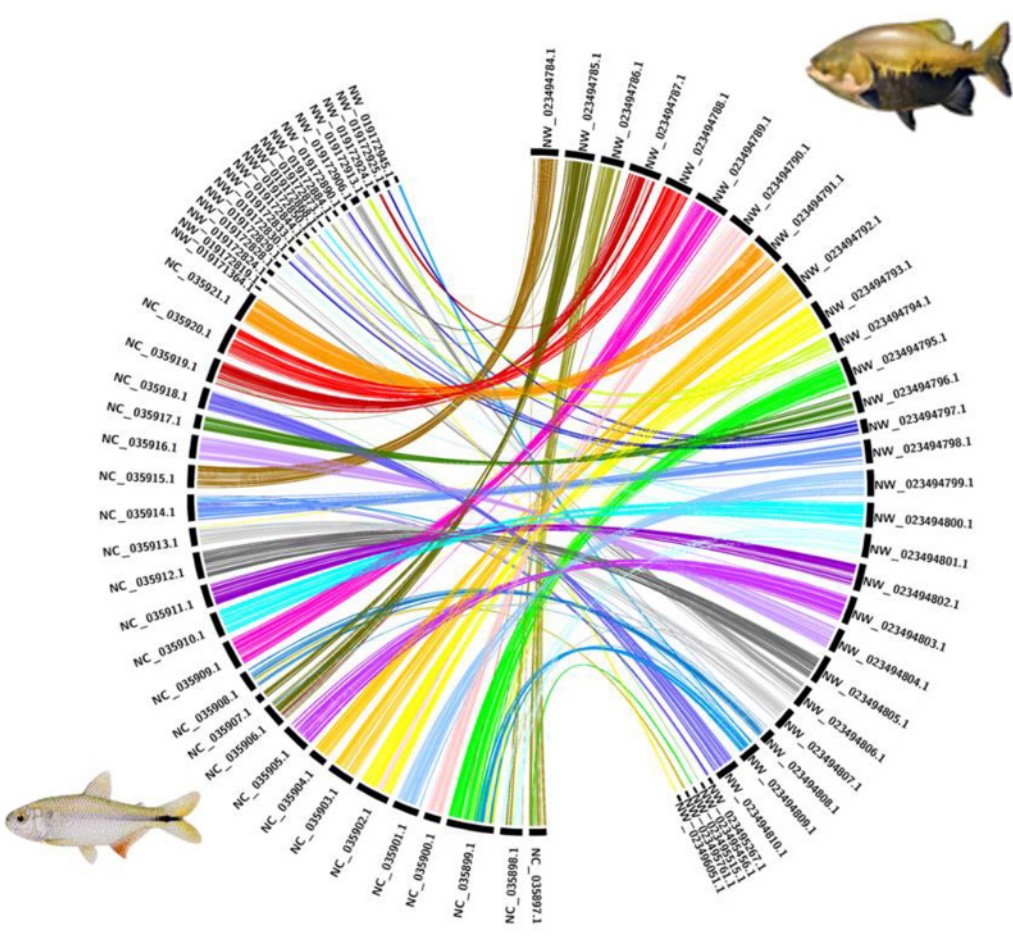

**Figure 6.** BUSCO gene synteny of *Colossoma macropomum* (tambaqui; right) and Astyanax mexicanus (Mexican tetra; left). Synteny analysis confirms moderate conservation of homologous single copy gene order between these species. While large syntenic blocks persist, a relatively large portion of *C. macropomum* genes are also fragmented into two or more linkage groups of the *A. mexicanus* genome.

## Re-use Potential

Seasonal and long-term modifications in environmental conditions are well-known to be associated with periodic events of low water dissolved oxygen leading to hypoxia and even anoxia. Tambaqui is an Amazon fish species that has developed adaptions to deal with this, such as enlargement of the lower lip to grasp oxygen better to survive in hypoxic conditions. These, along with other fish adaptations to the Amazon aquatic ecosystem, are intriguing scientific questions that could be scientifically addressed using the present well-assembled and annotated tambaqui genome. Moreover, the availability of this annotated genome will pave the way for the development of tools for genomic breeding programs of tambaqui, the most important native species for aquaculture in South America.

## DATA AVAILABILITY

The data sets supporting the results of this article are available in the *GigaScience* Database [55].



**Figure 7.** Whole genome-predicted single copy orthologs phylogeny of 12 fish species, including the newly sequenced genome of *Colossoma macropomum*. To the right of the phylogeny bars shows the proportion of different types of orthologs assigned by Orthofinder for each species.

All sequencing data is available on NCBI under the BioProjects PRJNA702552 and PRJEB40318. The former contains the Sequence Read Archive (SRA) experiments with accession numbers SRX10122091 to SRX10122101. The latter comprises the assembled genome and sequence annotations with the accession number GCF_904425465.1.

The genome sequence and annotation files—including coding sequences and proteins—can be downloaded from the NCBI FTP server [37]. A data viewer is also available [38].

## DECLARATIONS
## LIST OF ABBREVIATIONS

BUSCO: Benchmarking Universal Single-Copy Ortholog; Gbp: gigabase pair(s); Mbp: megabase pair(s); QV: consensus quality value.

## ETHICAL APPROVAL

We followed the applicable international and national ethical guidelines for the care and use of animals in research. The approval of the Ethics Committee for the Use of Animal registration is placed at the University of Mogi das Cruzes and is numbered #019/2017.

## COMPETING INTERESTS

The authors declare that they have no competing interests.

## FUNDING

The authors acknowledge FAPESP (São Paulo Research Foundation #2015/23883-0), National Council for Scientific and Technological Development (CNPq), and Coordination of Superior Level Staff Improvement (CAPES) through Tambaqui Project (Edital Pró-Amazonia – 047/2012) for financial support. AWSH, LLC, VWDAV, and DP are recipients of CNPq productivity scholarships (304662/2017-8, 304353/2019-1, 306718/2019-7, and 312273/2017-7, respectively).

## AUTHORS' CONTRIBUTIONS

AWSH, LLC, and DP designed and conceived this work; AWSH collected the samples; AWSH, MUS, DP, LLC, VMDAV wrote the manuscript; MUS and HM coordinated and carries out the bioinformatics analyses; AWSH, LLC and DP revised the manuscript. All authors read and approved the final manuscript.

## ACKNOWLEDGEMENTS

We acknowledge CEPTA-ICMBio (Centro Nacional de Pesquisa e Conservação da Biodiversidade Aquática Continental) for tambaqui individual contributions to this work, and Dr. Andrew Veale for critical review and language editing.

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
