## [Reviewer Report]

Comments on revised manuscriptThe revised manuscript (DRR-202107-01-R01) submitted by Hilsdorf et al. and accompanying answers fully answered my questions, and I am satisfied with these revisions. I have only find some minor errors should be corrected and one suggestion.
Minor erres:
In line 70: “ichthyo diversity” should be revised to “ichthyo-diversity” or “ichthyodiversity”.
In line 128: “BBBTools 37.90”. I guess this might be a typo error and it should be “BBTools”.
In line 133: I think the genome size is not “1,16 Gb” and is “1.16 Gb”, and a point should be used instead of comma.
One suggestion:
The Figure 5(a) includes genological time scale under the phylogenetic tree, but no corresponding descriptions in the main text. I think it is a good option for authors to include how this time tree was obtained and prepared in the main text.

---

## [Reviewer Report]

Reviewer name and names of any other individual's who aided in reviewer Zhong LiDo you understand and agree to our policy of having open and named reviews, and having your review included with the published papers. (If no, please inform the editor that you cannot review this manuscript.)YesIs the language of sufficient quality?YesPlease add additional comments on language quality to clarify if needed
Are all data available and do they match the descriptions in the paper? YesAdditional CommentsAre the data and metadata consistent with relevant minimum information or reporting standards? See GigaDB checklists for examples <a href="http://gigadb.org/site/guide" target="_blank">http://gigadb.org/site/guide</a>YesAdditional CommentsIs the data acquisition clear, complete and methodologically sound?YesAdditional CommentsIs there sufficient detail in the methods and data-processing steps to allow reproduction?YesAdditional CommentsIs there sufficient data validation and statistical analyses of data quality? YesAdditional CommentsIs the validation suitable for this type of data?YesAdditional CommentsIs there sufficient information for others to reuse this dataset or integrate it with other data?YesAdditional CommentsAny Additional Overall Comments to the AuthorThe manuscript “Genome assembly and annotation of the tambaqui (Colossoma macropomum): an emblematic fish of the Amazon River basin” by Alexandre et al. report a high-quality genome assembly of tambaqui (Colossoma macropomum) using different sequencing methods. 
Considering this species is valuable aquaculture species, the high quality-reference genome would provide great help for future breeding works.
This is an excellent work, and I only have a couple of comments related to the clarity and consistency of presentation of the genome. 

1. In line 40, “combined Illumina, PacBio and Hi-C sequencing technologies to generate a reference genome…”. 
However, no corresponding descriptions about Hi-C was found in the “Sequencing and assembly” and “RESULTS AND DISCUSSION” sections. As no Hi-C reads was found in PRJNA702552, this sentence should be corrected accordingly. 
2. In line 41 and 135~136, “assembled with 39X coverage of long reads and polished to a QV=36 with 169X coverage of short reads.” 
As displayed in Table 1, the total 169X coverage seem to the sum of all four libraries (including short reads, mate pair reads, and CLR reads) instead of short reads. 
3. In line 198 to 200, Alexandre et al performed synteny analysis using the first 54 scaffolds of C. macropomum against the BUSCO genes on the largest 50 scaffolds of the goldfish Carassius (ASM336829v1). However, only 32 scaffolds of C. macropomum (NW) were found in Figure 4. Does this mean no conserved synteny was found no the other 22 scaffolds of tambaqui?
4. In the “RESULTS AND DISCUSSION”, the author displayed a synteny analysis between tambaqui and goldfish Carassius auratus (line 198 ~ 201). I am wondering why not compare to Ictalurus punctatus (doi: 10.1038/ncomms11757 (2016)) and Astyanax mexicanus (https://doi.org/10.1038/s41467-021-21733-z), whose chromosome assembly are available now and have more close relationship.
5. Lastly, the language should be polished further.


RecommendationMajor Revision

---

## [Reviewer Report]

Reviewer name and names of any other individual's who aided in reviewer Filipe CastroDo you understand and agree to our policy of having open and named reviews, and having your review included with the published papers. (If no, please inform the editor that you cannot review this manuscript.)YesIs the language of sufficient quality?YesPlease add additional comments on language quality to clarify if needed
Are all data available and do they match the descriptions in the paper? YesAdditional CommentsAre the data and metadata consistent with relevant minimum information or reporting standards? See GigaDB checklists for examples <a href="http://gigadb.org/site/guide" target="_blank">http://gigadb.org/site/guide</a>YesAdditional CommentsIs the data acquisition clear, complete and methodologically sound?YesAdditional CommentsIs there sufficient detail in the methods and data-processing steps to allow reproduction?YesAdditional CommentsIs there sufficient data validation and statistical analyses of data quality? YesAdditional CommentsIs the validation suitable for this type of data?YesAdditional CommentsIs there sufficient information for others to reuse this dataset or integrate it with other data?YesAdditional CommentsAny Additional Overall Comments to the AuthorThe manuscript by Hilsdorf and colleagues is of the utmost importance for those working with tambaqui, a valuable species in many respects. The quality of the assembly is excellent as is these initial analysis to validate their assembly. I am sure it will assist those that work in the biology of this species. Thus, the work should be accepted. I have very few comments, except to note to the authors that additional RNA seq projects and a preliminary genome assembly (much poorer than the present one) have been published and show the value of these approaches particularly in the context of tambaqui aquaculture (lines 84, 87 and 95) . These should probably be referenced? For example, tambaqui dietary requirements of long chain PUFA (a fundamental aspect of nutrigenomics of teleost fish) has been molecularly studied through the isolation and functional characterization of genes (e.g. elovl2, fads2, Elovl4a and Elovl4b) that were identified in these initial omic datasets.RecommendationAccept